# Development of Approaches for Transgene Expression in the Pathogenic Free-Living Amoeba *Naegleria fowleri*

**DOI:** 10.3390/pathogens15010012

**Published:** 2025-12-22

**Authors:** Caroline M. Palmentiero, Jillian E. M. McKeon, Colm P. Roster, James C. Morris

**Affiliations:** Eukaryotic Pathogens Innovation Center, Department of Genetics and Biochemistry, Clemson University, Clemson, SC 29634, USA; cpalmen@g.clemson.edu (C.M.P.); jmilane@clemson.edu (J.E.M.M.); croster@clemson.edu (C.P.R.)

**Keywords:** transgene, transgenic, *Naegleria fowleri*, fluorescent proteins, selectable markers

## Abstract

The absence of molecular tools for manipulation of gene expression in the pathogenic free-living amoeba *Naegleria fowleri* has historically limited our understanding of gene function in the organism and has coincidently impacted the identification of potential druggable pathways and proteins. Here, we describe the development of approaches for the generation of transgenic amoebae using polyethyleneimine nanoparticles to deliver plasmids designed to confer antibiotic resistance and fluorescence to the cells. Through a series of optimization steps, we found that transfection of plasmids encoding the fluorescent protein mCherry fused by a T2A self-cleaving peptide to a codon-optimized puromycin acetyltransferase selectable marker yielded fluorescent cells that were resistant up to 100 µg/mL puromycin. Transfected trophozoites harbored between 45 and 65 copies of the transgene per cell and both fluorescence and resistance were persistent in the presence of selector through continued passages. The development of these approaches is anticipated to enable application of an array of genetic manipulation techniques including forward and reverse genetics to the study of this important pathogen.

## 1. Introduction

Human infection by the pathogenic free-living amoeba *Naegleria fowleri* leads to primary amoebic meningoencephalitis (PAM). PAM is a life-threatening illness that occurs when the amoeba invades the brain following introduction of contaminated water into the nasal passages. Tragically, the vast majority (>95%) of infections end in fatality [1]. While perceived as rare (with 167 cases in the US from 1962–2024, CDC), case numbers from developing parts of the world are likely under-reported because of a lack of postmortem evaluation [2]. Major hurdles for improved patient outcomes include early and efficient diagnosis approaches, currently being worked on by the CDC and others, and the paucity of efficacious therapeutic options.

A major impediment in the development of new treatments is the lack of robust and reliable genetic tools for studying the amoeba. These tools are critical for drug discovery in two ways. First, genetic approaches provide a means to understand fundamental pathogen biology, allowing the identification of key pathways or processes for targeting with new therapies. Second, without these tools, it is not possible to genetically determine if a potential protein target is essential, a key early validation requirement for progression through target-based drug discovery pipelines. Further, it is not possible to establish chemical-genetic validation by engineering in resistant alleles of a target gene, the highest form of validation used to confirm that an agent exerts its lethal impact through a particular target.

The evidence of molecular manipulation of *N. fowleri* is scant, and the approaches used to generate these tools are not clear [3,4]. However, genetic tools have been described for use in the non-pathogenic *Naegleria gruberi*. Using electroporation, *N. gruberi* was stably transfected with an ectopic vector that allowed antibiotic resistance selection and fluorescent protein gene expression [5]. Other pathogenic amoebae, including *Acanthamoeba* and *Entamoeba*, have been successfully modified to express transgenes, supporting the possibility of generating approaches for transgenesis in *N. fowleri*.

Here, we report the development of a plasmid-based approach for transgene expression in *N. fowleri*. We tested a diversity of gene untranslated regions (UTRs) in the search for useful promoter and other regulatory sequences. Additionally, we assessed the utility of available antibiotic resistance genes for use as selectable markers and incorporated these into a variety of designs. These were subject to a diversity of transfection approaches, with the best current approach including the use of nanoparticle-associated plasmids encoding a fluorescent protein reporter fused to a selectable marker protein by a T2A self-cleaving peptide. Constructs using this format generated long-term resistance and fluorescence.

## 2. Materials and Methods

Amoebae Culturing—*N. fowleri* TY strain (NfTY, ATCC 30170, generously provided by Dr. Dennis Kyle (University of Georgia)) trophozoites were cultured at 37 °C in Nelson’s Complete Media (NCM; 0.17% liver infusion broth, 0.17% glucose, 0.012% sodium chloride, 0.0136% potassium phosphate monobasic, 0.0142% sodium phosphate dibasic, 0.0004% calcium chloride, 0.0002% magnesium sulfate, 10% heat-inactivated fetal bovine serum, 1% penicillin-streptomycin) in treated tissue culture flasks. For cell passage, cells were placed on ice for 20–30 min, centrifuged (3000× *g*, 4 °C, 5 min), resuspended in 1 mL NCM, and passed into fresh media.

Plasmid Design and Cell Transformation—The plasmids described here are built on one of two bacterial backbones. The early generation (.v1 and .v2) plasmids shared 636 bp fragment of the *N. gruberi* circular extrachromosomal ribosomal element (CERE) that is known to contain the origin of replication of that vector [6,7]. For .v4 plasmids and later, the backbone was a 2328 bp fragment from a bacterial cloning vector. pPACeYFP.v1 and pPACmCherry.v1 included a 1090 bp portion of the 5’ UTR (from −1091 to −1) and a 349 bp fragment of the 3’ UTR (+1–349 from the stop codon) of a putative actin gene (NfTy_067520, amoebaDB.org) flanking the PAC gene and a 1090 bp portion of 5’UTR (from −1091 to −1) and a 1090 bp fragment of the 3’UTR (+1–1090 from the stop codon) of a putative ubiquitin gene (NfTy_087590) flanking the fluorochrome. In .v2 plasmids, the 5’ actin UTR upstream of PAC was replaced with the 5’ UTR of ubiquitin. The .v4 plasmid had the same architecture as the .v1 plasmids but included codon-optimized PAC and eYFP. (Please note that during development of pPACeYFP.v4, a 38 bp fragment of the 5’ ubiquitin UTR (from −1090 to −1052) was deleted). The vectors .v5 and .v6 included the actin 5’ and 3’ regions flanking both transgenes. The .v7 and .v8 included the actin 5’ and 3’ UTRs flanking either codon optimized eYFP or mCherry fused to codon optimized PAC by the T2A self-cleaving peptide sequence. (Sequences for backbones and inserts are provided in Appendix A. See Appendix A for a schematic with features of pPACmCherry.v8.)

For transfection, 10^4^ cells were inoculated into 2 mL of NCM in a treated 6-well plate and cultured for 24 h (37 °C, 5% CO_2_). PEI-DNA nanoparticles were assembled 30 min before transfection. For a single transfection (per well on the 6-well plate) PEI (Kyfora Bio, Horsham, PA, USA, 24765100) was added to a final volume of 50 µL Opti-MEM media (Gibco, 31985-070) in a sufficient quantity to yield a final 1:1 *w*/*w* ratio with the DNA. The plasmid (typically 5–10 µg) was added to a final volume of 50 µL in Opti-MEM media and then gently mixed with the PEI solution. After incubation (RT, 20 min), Opti-MEM media (400 µL) was gently mixed into the solution. Media was removed from the 6-well cultures and replaced by dropwise introduction of the PEI-DNA solution, gently rotating the well plate to ensure coating of the bottom of the well. Cells and PEI-DNA solution were incubated for 4–5 h (37 °C, 5% CO_2_), NCM (1 mL) was added to each well, and incubated for 18 h (37 °C, 5% CO_2_). The PEI-DNA mixture was removed and NCM (2 mL) added to the culture. Selection was applied by addition of fresh NCM harboring the antibiotic 48 h after transfection.

Cell Imaging—Live cells were imaged using a Leica DMi8 epifluorescence microscope. Fluorescent cells were detected using either an eYFP excitation/emission filter set (490–510 nm/520–550 nm) or a Texas Red excitation/emission filter set (540–580 nm/592–668 nm). Images and videos of live cells were captured throughout the transfection timeline, beginning at 48 h post-transfection and processed using the LasX software (Leica Application Suite X, 3.10.1.29575).

Plasmid Copy Number Determination and Western Blotting—Ectopic vector concentration was quantified by Nanodrop Lite spectrophotometry (ThermoFisher, Waltham, MA, USA) and copy number/μL was calculated using the ThermoFisher DNA Copy Number and Dilution Calculator based on the 5144 bp size of pPACmCherry.v8. A qPCR standard curve was generated using five-fold dilutions from 9.01 × 10^8^ to 1.15 × 10^4^ copies of pPACmCherry.v8. Reactions were performed in triplicate using Luna qPCR Master Mix (New England Biolabs, Ipswich, MA, USA) according to the manufacturer’s specifications on a QuantStudio 3 Real-Time PCR instrument (Applied Biosystems, Waltham, MA, USA). The average Cq value was plotted on the y-axis with the corresponding copy number on the x-axis. Genomic DNA from *N. fowleri* trophozoites (1 × 10^6^ cells) was extracted using the DNeasy Blood and Tissue Kits (Qiagen, Hilden, Germany) and DNA from 25,000 cell equivalents was used as template for each reaction. The two amplified targets were the bacterial backbone within the plasmid and the PAC gene. Cq values were then compared to the standard curve to determine copy number per cell equivalent. (Please see Appendix A for primer sequences and Appendix A for standard curves.)

For Western blotting, cells transfected with pPACmCherry.v8 and cultured in puromycin for 51 days were harvested by centrifugation (3000× *g*, 4 °C, 5 min) and pellets were resuspended in sample cracking buffer prior to heating (5 min, 95 °C). Proteins were resolved on a 15% SDS polyacrylamide gel and transferred to a PVDF blotting membrane (Cytiva, Marlborough, MA, USA) for Western blotting using rabbit anti-mCherry primary antibody (16D7, 1:10,000), and goat anti-rabbit HRP secondary antibody (31460, 1:10,000) (ThermoFisher Scientific, Waltham MA, USA). Western blots were visualized using an iBright (ThermoFisher Scientific, Waltham, MA, USA).

## 3. Results

### 3.1. Identification of Useful Antibiotic Selectable Markers

The ability to select against untransfected cells using antibiotics is critical for the development of a tool that is widely accessible. We considered standard antibiotics that are generally active against eukaryotes, including blasticidin, hygromycin, phleomycin, and puromycin. First, we assessed untransfected trophozoite sensitivity to the agents, as an ideal selection reagent would not be needed at high doses, thereby reducing the cost of the experiment. Blasticidin was the most potent anti-amebic, with an EC_50_ value of 0.39 ± 0.01 µg/mL, while puromycin, phleomycin, and hygromycin were less potent (EC_50_ values of 3.84 ± 0.20 µg/mL, 8.79 ± 0.23 µg/mL, and 94.9 ± 14.2 µg/mL, respectively, Appendix A). Because we were uncertain about the level of resistance transfectants might have, we arbitrarily assessed the impact of the antibiotics on untransfected cells at a variety of antibiotic concentrations. Notably, blasticidin never eliminated all the trophozoites at any of the concentrations we tested. However, at five times the EC_50_ concentration, puromycin eliminated all untransfected trophozoites, so we pursued this antibiotic as our selectable marker.

### 3.2. Polyethyleneimine (PEI) Nanoparticle-Mediated Transfection Generates Stable Transgenic Cell Populations

Because optimized transfection conditions for *N. fowleri* have not been described, we initially used approaches that were developed for *N. gruberi* transfection, using electroporation to deliver an ectopic vector [5]. Using electroporation of various plasmids at different concentrations (described below, see Figure 1 for description), we were unable to detect successful transfection as scored by the presence of fluorescent cells at 48 h after electroporation or by resistance to puromycin (at 20 µg/mL) after one week of culturing. Therefore, we did not pursue further analysis.

Additional efforts included using routine transfection approaches like lipofectamine-based reagents (we tested SuperFect and ViaFect Transfection Reagents (Promega, Fitchburg, WI, USA)) and nucleofection using a Nucleofector II System with Human T-cell Nucleofector solution and Programs X-001 and X-29 (Amaxa, Cologne, Germany). None of these resulted in measurable transfections, leading us to consider alternative transfection approaches used successfully in other amoebae.

Recently, Moreno et al. described using polyethyleneimine (PEI)-based nanoparticles for transfection of *Acanthamoeba castellanii* [8]. We tested two PEIs, PEI-25 and PEI-40, that differ based on their size (25 kDa and 40 kDa, respectively). By gel electrophoresis, we found that a ratio of 1:1 plasmid DNA to PEI-40 resulted in retention in the gel wells, a measure of incorporation of the DNA into large complexes. We moved forward with this amount for future studies. Please note that controls of PEI alone in transfections were not used, as the PEI does not form particles in the absence of the organizing nucleic acid; instead, our control transfections for scoring general toxicity of the transfection typically contained a bacterial plasmid (pGEM-T) bearing a gene unrelated to this work organized into particles. Using this reagent associated with pPACeYFP.v1, we note transfected cells were faintly fluorescent 48 h after DNA delivery (Figure 2A), suggesting that PEI nanoparticles could provide a suitable means of exogenous plasmid delivery.

### 3.3. Improvements to the Vector

Our vectors were designed based on expression vectors used in other amoebae [5]. The plasmid pPACeYFP.v1 included the puromycin resistance gene (PAC) flanked by 5’ and 3’ untranslated regions from a putative actin gene (Figure 1). Additionally, the plasmid included eYFP flanked by 5’ and 3’ untranslated regions from a putative ubiquitin gene. Transfection of this plasmid yielded our first hints of success, with ~1 in 1000 transfected cells being fluorescent 48 h after transfection (Figure 2A). Application of puromycin (20 µg/mL) at that point eliminated the transfected amoebae. (Please note, we attempted gradually introducing puromycin, initiating treatment at 48 h at 5 µg/mL and then increasing to 20 µg/mL over several days, at which point control and plasmid-transfected cells were all dead). Similar results were observed when we replaced the eYFP gene for mCherry (pPACmCherry.v1), with fluorescence detected in cells prior to application of the selection antibiotic, at which point all cells were killed (See Figure 2B for representative cells).

These observations suggested that the ubiquitin UTRs were capable of transiently supporting transgene expression, while the utility of the actin UTRs was unclear. Therefore, we replaced the actin 5’ region upstream of the PAC gene with the same ubiquitin 5’UTR that had generated visible fluorochrome expression. Neither of the plasmids that had this modification, pPACeYFP.v2 or pPACmCherry.v2, conferred puromycin resistance to recipient cells, indicating that expression issues might have resulted from problems with the PAC protein coding sequence. To explore this, we replaced the eYFP and PAC genes with versions optimized for *N. fowleri* expression, yielding pPACeYFP.v4. The codon-optimized versions of the ORFs were generated using the Codon Usage Database (https://www.kazusa.or.jp/codon/, accessed on 11 November 2024) [9]. (Please note in the production of this construct a 38 bp fragment of the 5’ ubiquitin UTR (from 1052–1090 bp) was deleted, yielding a truncated ubiquitin 5’ UTR).

Amoebae were transfected with pPACeYFP.v4 and puromycin was added at 48 h to initiate selection. At that point, ~10% of the cells expressed visibly detectable fluorescence (Figure 3A). Resistant cells grew from the culture, dividing normally in the presence of 20 µg/mL puromycin. Using PCR, we detected PAC in genomic DNA preparations on day 12 from two different clonal populations (Appendix A). However, the percentage of fluorescent cells did not increase with the generation of resistance. The few cells that remained eYFP positive were unusually elongated and the fluorophore expression was not evenly distributed throughout the cytoplasm but instead was primarily localized into vesicle-like puncta in the amoeba (Figure 3A). Replacing the ubiquitin UTRs flanking the codon-optimized fluorescent protein with the actin UTRs (to yield pPACeYFP.v5) did not improve fluorescence expression.

The limited eYFP expression and subcellular organization of the fluorochrome into puncta suggested the protein was not well tolerated by the amoeba. To address this, we replaced the eYFP with mCherry, a more photostable fluorochrome that shares limited (~30%) sequence identity with the eYFP (pPACmCherry.v6). Again, fluorescence was noted in a limited portion of the cells, but this fluorescence was transient (Figure 3B).

Our inability to drive expression of sufficient fluorescent protein using the same UTRs that yield puromycin resistance from PAC expression (in the same vector, pPACeYFP.v5 for example) led us to explore methods to connect expression of resistance to the expression of fluorescence. In pPACeYFP.v7, the eYFP and PAC ORFs were separated by a T2A self-cleaving peptide sequence, with the entire fusion construct under the regulation of the actin 5’ and 3’ UTRs. This approach yields a single transcript harboring both ORFs, thereby forcing the cell to generate a transcript to eYFP to express PAC. During translation, the ribosome skips on the T2A sequence, resulting in a failure to form a peptide bond between the two ORFs, yielding two independent proteins [10]. Using this approach, we generated puromycin resistant amoebae with apparent cytosolic eYFP expression in ~40% of the transfected cells that were otherwise phenotypically normal (Figure 4A). Unfortunately, we were unable to increase the percentage of fluorescent cells by either limiting cloning or fluorescent cell sorting, suggesting the intensity of fluorescence was not heritable but rather the result of differences in protein stability in the population.

Our initial efforts with mCherry suggested it was potentially better tolerated by the amoebae, so we also developed a vector harboring the fluorochrome fused to PAC with a T2A self-cleaving peptide (pPACmCherry.v8). (Please note, the mCherry was not codon optimized). Following puromycin selection, we found that ~70% of the cells were visibly fluorescent (Figure 4B). Analysis of the transgene copy number by qPCR suggests that, on average, each cell equivalent had 45–65 copies of the DNA. Further, increasing the selection pressure to 100 µg/mL increased the relative fluorescent intensity and percentage of visibly fluorescent cells (Figure 4B). Expression was confirmed by Western blots of these cells after 52 days of selection (Appendix A). Cells recovered from long-term storage at −80 °C remained fluorescent, indicating that the transgene did not alter the tolerance of the amoebae to cryopreservation.

## 4. Discussion

Infection with *N. fowleri* is almost invariably (>95%) lethal. The challenges faced in overcoming the dire prognosis associated with the infection are two-fold. First, earlier detection is likely important to improve outcomes. Confounding successful treatment is the lack of efficacious therapeutic options, an issue that will remain until new drugs are developed. Experiments that identify unique and essential parasite biology are key steps in the development of new target-based therapeutics. Here we have described the development of genetic tools that will, for the first time, allow application of genetic manipulation to the study of *N. fowleri*.

Genetic tools will power both basic discovery and genetic validation of drug targets, processes that will revolutionize our understanding of the biology behind infection. Currently, there is a lack of fundamental understanding of gene function in the amoebae, which largely results from our inability to manipulate gene expression and perform basic genetic experiments. This deficiency impairs our ability to understand the biochemistry and cell biology of the organism—areas key to identifying potential targets for therapeutic intervention. Additionally, target-based drug discovery campaigns are severely hampered by a lack of methods for genetic validation of targets. We and others have identified promising pathways and associated candidate proteins that could be targets for development, with many of these potential targets being attractive given resolution of their structures by X-ray crystallography [11,12]. For example, we have found that glycolysis is critical to the pathogen viability using chemical inhibitors of the pathway [11]. While the first enzyme in glycolysis, glucokinase, has properties that distinguish it from human host enzymes and its structure has been solved, our inability to genetically validate this target tempers our pursuit of inhibitors as lead drug compounds. Recent efforts from the Rice/Kyle group screening the Calibr ReFRAME library identified additional chemically implicated targets that now need tests of essentiality and chemical-genetic validation [13].

The tools previously described for *Naegleria* spp. include ectopic vectors designed for use in *N. gruberi*, a non-pathogenic species [5]. This system included putative *N. gruberi* gene regulatory sequences flanking antibiotic resistance and fluorescent protein genes and was introduced into the amoebae by electroporation. We were unable to generate transfected *N. fowleri* using a very similar vector that harbored the same *N. gruberi* regulatory sequences. This is perhaps not surprising given the differences in the genomes of the organisms. The *N. fowleri* genome (genome size ~29.6 Mbp for the reference strain) generally shares limited coding sequence similarity with *N. gruberi* (32.1% of the ~17,000 predicted ORFs aligned to the *N. gruberi* genome) [14], and assembly statistics from two additional *N. fowleri* strains indicate marked differences between the pathogenic species and *N. gruberi* [15]. Other tools have been described in *Naegleria*, with RNAi used to knock down *nfa1* in *N. fowleri* trophozoites [3,4]. We have not yet been able to replicate this approach, suggesting potential strain-dependent differences.

In our eYFP transfectants, the limited percentage of visibly fluorescent cells and the organization of the fluorochrome in those cells into puncta suggested it was not well tolerated by the amoeba. While this could result from multiple factors, it is possible that the transgene expression depleted rare charged tRNAs, a consequence that could be addressed by exogenous co-expression of tRNAs as is done in Rosetta bacteria (Novagen). Alternatively, at highly transcribed levels, protein may be toxic, overwhelming the cellular protein folding machinery and triggering the formation of aggregates that could be detrimental to the organism.

We have not been able to re-isolate free plasmid from the amoebae using standard mini-prep approaches. However, wristwatch PCR [16] on genomic DNA extracted from pPACeYFP.v7 transfected cells suggests that at least a portion of the transfected plasmid remained extrachromosomal in the cell. Using PAC primers with wristwatch PCR primers in sequential combination generated a PCR product that included a portion of PAC and the actin 3’UTR, along with part of the bacterial origin, the aatB2 site, the SV40 signal, and the ori sequence of the pPACeYFP.v7 backbone. (See Appendix A for the complete sequence). Notably, the sequence lacked genomic DNA that would have suggested integration into the genome, so it is possible the plasmid remains an episome or has been linearized but remains extrachromosomal.

Expression of fluorochromes serves as a proof-of-concept for future efforts. These include expression of genes of interest that are tagged with antibody epitopes or fluorescent proteins. Additionally, we anticipate appending subcellular localization sequences (peroxisomal targeting or nuclear localization sequences, for example) on fluorescent proteins to label organelles of interest. The organellar label could then be used to monitor purification of the organelle. We also envision expressing gene editing or gene silencing components from the construct, including Cas9 and Cas13. In the case of Cas9, the editing protein would form the ribonucleoprotein complex after transfection of exogenously generated guide RNAs. Repair templates could also be introduced to engineer in selectable markers or tags into the targeted allele. This sort of approach, using a library of gRNAs, could be used to generate a library for the identification of every essential gene during infection, expanding the list of putative drug targets. Cas13, which impacts mRNAs, could be used to silence gene expression, enabling RNAi-like studies in the amoeba. Last, the tool may enable in vivo imaging during infection, allowing for a chronicling of the infection course through time. We found that fluorescence from pPACeYFP.v7 and pPACmCherry.v8 persisted in the absence of selection for about one week, suggesting that expression of a near-infrared reporter or luciferase might be sufficient for IVIS-type imaging in infected rodents without continued selective pressure.

## Figures and Tables

**Figure 1 pathogens-15-00012-f001:**
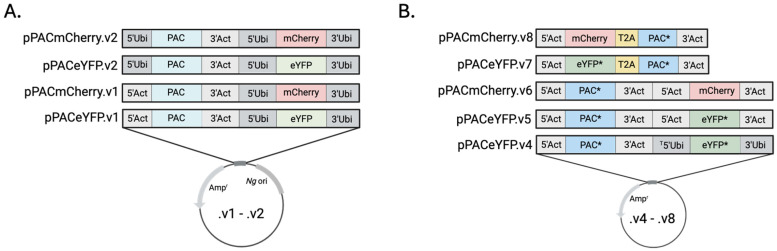
Schematic of expression vectors used for transgene expression. Inserts were cloned into a vector backbone from VectorBuilder (pRP[Exp]). (Appendix A for the backbone sequence). (**A**) Plasmids .v1–.v2 harbor the *N. gruberi* CERE origin. (**B**) Plasmids .v4–.v8. * Codon optimized. ^T^5’Ubi indicates a truncation on the 5’ end of Ubiquitin 5’UTR. Amp^r^ is the ampicillian resistnace gene. Image created using BioRender.com.

**Figure 2 pathogens-15-00012-f002:**
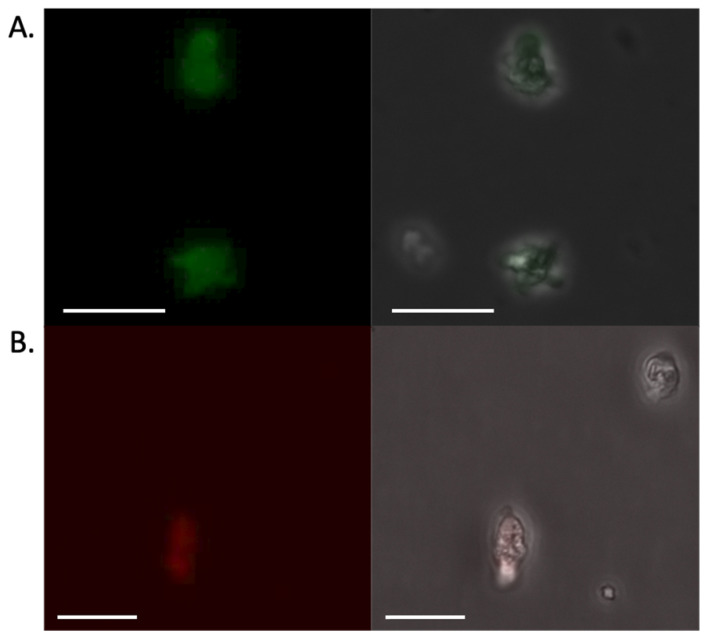
Transfection with pPACeYFP.v1 yielded faint, transient fluorescence. (**A**) *N. fowleri* trophozoites 72 h post-transfection. Cells were transfected with 5 µg pPACeYFP.v1 using PEI-40 and selected with 20 µg/mL puromycin 48 h after transfection. (**B**) Trophozoites 48 h post-transfection. Cells were transfected with 5 µg pPACmCherry.v2. Images were captured on a Leica DMi8 inverted microscope using the eYFP excitation/emission filter set (490–510 nm/520–550 nm) for (**A**) and the Texas Red excitation/emission filter set (540–580 nm/592–668 nm) for (**B**). Scale bars are 25 µm.

**Figure 3 pathogens-15-00012-f003:**
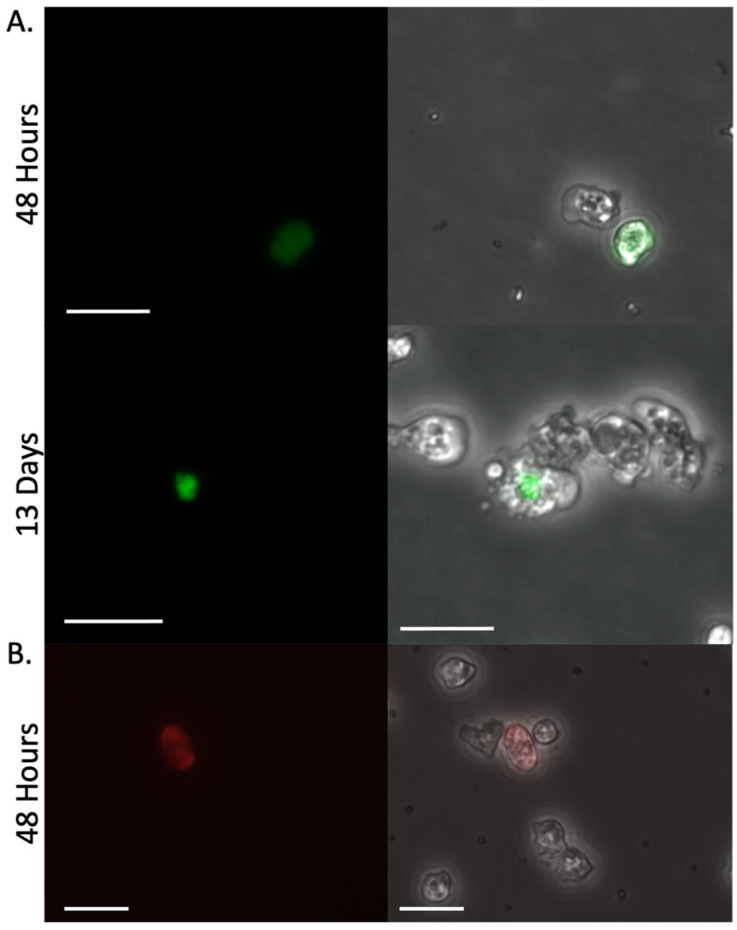
Transfection of *N. fowleri* cells with codon-optimized PAC and eYFP. (**A**) Trophozoites 48 h and 13 days after transfection with 10 µg pPACeYFP.v4 using PEI-40. Selection was initiated with 20 µg/mL puromycin at 48 h. (**B**) Amoebae 48 h post-transfection with 10 µg pPACmCherry.v6 using PEI-40 before initiation of selection. Please note, the pPACmCherry.v6 cells did not remain fluorescent after selection. Images were taken using the excitation and emission filter sets described in the Figure 2 legend. Scale bars are 25 µm.

**Figure 4 pathogens-15-00012-f004:**
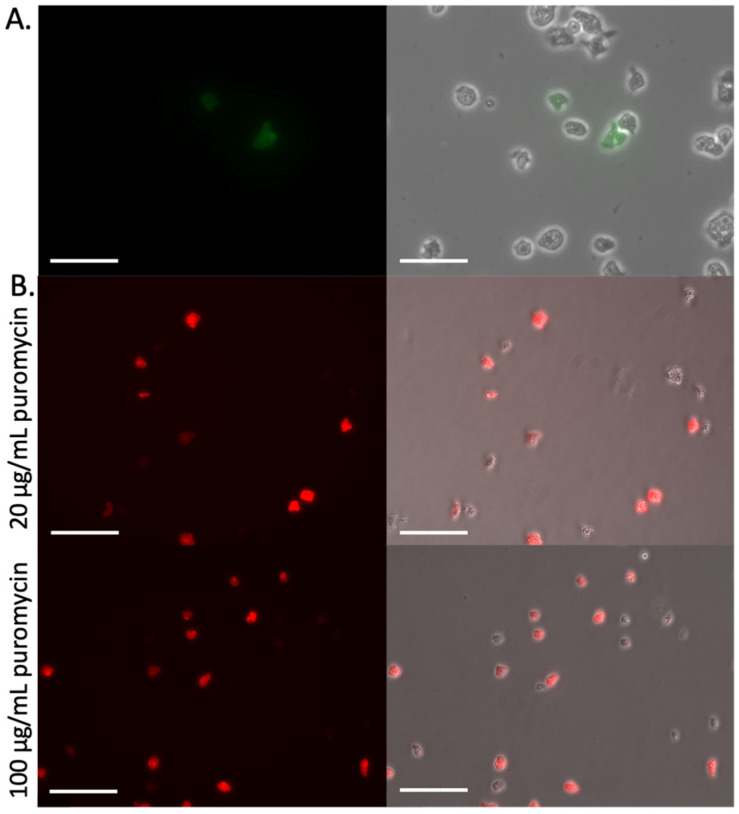
Inclusion of a T2A self-cleaving peptide resulted in robust fluorescence. (**A**) *N. fowleri* trophozoites 17 days post-transfection with 5 µg pPACeYFP.v7; selection had been initiated 48 h post-transfection with 20 µg/mL puromycin. Scale bars are 50 µm. (**B**) Trophozoites eight days after transfection with 5 µg pPACeYFP.v8. After 21 days, puromycin was increased to 100 µg/mL and trophozoites were imaged 48 h after the increase in pressure. Images were collected using the same excitation and emission filter sets described in the Figure 2 legend. Scale bars are 100 µm.

## Data Availability

The original contributions presented in this study are included in the article/Appendix A. Further inquiries can be directed to the corresponding author.

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
