# Peer review of "Development of Approaches for Transgene Expression in the Pathogenic Free-Living Amoeba *Naegleria fowleri"

_pathogens, 2025, doi:10.3390/pathogens15010012_

Round 1

Reviewer 1 Report

Comments and Suggestions for Authors

This manuscript reports on the development of plasmid vectors capable of expression of transcripts in the human pathogen Naegleria fowleri. This has been especially challenging as this amoeba is a member of an extremely divergent group of organisms that are very poorly characterised regarding genetic tools, and basic information such as antibiotic selectable markers and the efficiency of various transfection methods. Significant progress has been made and GFP based proteins have been successfully produced in these amoebae in genetically stable N. fowleri cell lines. This ability will greatly facilitate further research using forward and reverse genetics as the authors claim.

This manuscript is well written and very clear, a few very minor points are noted below.

This work has generated a large number of different (but related) plasmids. The general construction is outlined in figure 1 but perhaps the inclusion of a full diagram (map) of the most promising of these would help readers understand how these work.

The units for length are given as μM throughout the manuscript but μm is normally used.

Lines 149, 150, 163 (also more generally) ensure that the species names are italicised consistently.

Line 212 Amoebae rather than Amoeba.

Author Response

  1. This work has generated a large number of different (but related) plasmids. The general construction is outlined in figure 1 but perhaps the inclusion of a full diagram (map) of the most promising of these would help readers understand how these work.

Response:  We have now included as supplemental material Figure 1 the map of the most promising vector, the pPACmCherry.v8, as it is the most advanced plasmid and includes features developed in earlier plasmid versions.

  1. The units for length are given as μM throughout the manuscript but μm is normally used.

Response:  We have corrected this mistake.  We apologize for this typographical error.

  1. Lines 149, 150, 163 (also more generally) ensure that the species names are italicised consistently.

Response:  We have corrected the formatting as indicated.

  1. Line 212 Amoebae rather than Amoeba.

Response:  We have corrected the word to indicate plural amoebae. 

Reviewer 2 Report

Comments and Suggestions for Authors

In this study, researchers delineate the evolution of methodologies for the generation of transgenic amoebae, employing polyethyleneimine nanoparticles as a delivery mechanism for plasmids engineered to confer antibiotic resistance and fluorescent properties to the cells. Although, the manuscript was well written and provided new molecular tools for manipulation of gene expression in the pathogenic free- living amoeba Naegleria fowleri. Some corrections and revision should be considered before publication.

- Species names should be written in italics please revise all the MS to correct it, (line 149- 150)

- Please provide all the figures with higher dimensions to be much more legible,

- Please provide much more detail in figure captions (including the channel name and overlay).

- In all the figures, captions, authors have provided scale bars in µM (this is the usual international unit for Molarity, the international unit for meter should be written in minuscule µm). Please revise and correct accordingly.

Author Response

  1. Species names should be written in italics please revise all the MS to correct it, (line 149- 150)

Response:  We have corrected the formatting as indicated.

  1. Please provide all the figures with higher dimensions to be much more legible

Response:  We have enlarged and increased the resolution of text-bearing figures to make them more legible.  Additionally, we have included a map of pPACmCherry.v8 (the most promising vector) as a supplemental figure to provide additional details.

  1. Please provide much more detail in figure captions (including the channel name and overlay).

Response:  We have modified the figure captions to include channel names and additional information.

  1. In all the figures, captions, authors have provided scale bars in µM (this is the usual international unit for Molarity, the international unit for meter should be written in minuscule µm). Please revise and correct accordingly.

Response:  We have corrected this mistake.  We apologize for this typographical error.